# Soil Health Evaluation of Farmland Based on Functional Soil Management—A Case Study of Yixing City, Jiangsu Province, China

**Rui Zhao [1] and Kening Wu [1,2,3,*]**

[1] School of Land Science and Technology, China University of Geosciences, Beijing 100083, China; zhaoruifighting@cugb.edu.cn
[2] Key Laboratory of Land Consolidation and Rehabilitation, Ministry of Natural Resources, Beijing 100035, China
[3] Techanology Innovation Center of Land Engineering, Ministry of Natural Resources, Beijing 100035, China
* Correspondence: wukening@cugb.edu.cn

**Abstract:** Given that farmland serves as a strategic resource to ensure national food security, blind emphasis on the improvement of food production capacity can lead to soil overutilization and impair other soil functions. Hence, the evaluation of soil health (SH) should comprehensively take soil productivity and ecological environmental effects into account. In this study, five functions from the perspective of functional soil management were summarized, including primary productivity, provision and cycling of nutrients, the provision of functional and intrinsic biodiversity, water purification and regulation, and carbon sequestration and regulation. For each soil function, in view of the natural and ameliorable conditions affecting SH, basic indicators were selected from the two aspects of inherent and dynamic properties, and restrictive indicators were chosen considering the external properties or environmental elements, with the minimum limiting factor method coupled with weighted linear model. The new evaluation system was tested and verified in Yixing City, China. The healthy and optimally functional soils were concentrated in the northeast and mid-west of Yixing City, whereas unhealthy soils were predominant in the south and around Taihu Lake. The main limitations to SH improvement included cation exchange capacity, nutrient elements, and soluble carbon. The SH evaluation method was verified using the crop performance validation method, and a positive correlation was noted between food production stability index and soil health index, indicating that the evaluation system is reasonable.

**Keywords:** soil ecosystem services; soil health; soil multifunctionality; soil obstacles; sustainable soil management

## 1. Introduction

Among the 17 sustainable development goals covered by the United Nations (UN) Sustainable Development Goals (SDGs), 13 goals are directly or indirectly related to soil, the basis for sustainable food production and environmental protection [1]. Soil plays a key role in agriculture and is the medium in which nearly all food-producing plants grow. Soil degradation and pollution affect food security and pose a serious threat to human survival [2,3]. At present, human society is facing the challenges of global population, resources, environment, ecology, and other issues, which have increased the burden on global soil resources as a result of population pressure and related land-use changes, exerting a tremendous impact on food security and the sustainable development of agriculture [4–7]. In addition, the sudden global public health crisis in 2020, the COVID-19 pandemic, had a severe impact on the food supply system. Soils provide nutritious food and other products as well as clean water and flourishing habitats for diverse organisms. Around the world, the agricultural sector has been facing the challenge of increasing primary productivity to

meet the growing global food security needs [8], and the difficulty of land use management and agricultural environmental protection has been increasing. The new research and innovation program of the European Union (EU) for the period of 2021–2027, Horizon Europe, has defined five mission areas, including SH and food, thus recognizing the importance of soils for sustainable development [9]. Recently, under the promotion of the United Nations Food and Agriculture Organization, International Soil Science Federation, and other institutions, a consensus has been reached on the concept of SH management and protection; however, a clear method for assessing SH has not yet been defined.

As the most populous country in the world, China has a particularly urgent need for food security, and ensuring food production has become an important part of the national economy. China has a small per capita farmland area of only one-third of the global average level, with some areas presenting low soil quality (SQ) and serious soil degradation [10]. According to the "National Survey Bulletin of Soil Pollution Status", the over-standard rate of soil pollution in China's farmland is as high as 19.4%, with main pollutants such as cadmium (Cd), nickel (Ni), copper (Cu), zinc (Zn), and mercury (Hg). The soil of the major grain production area in the south of China is more seriously polluted with heavy metals than that in the north, imposing a severe threat on safe utilization of farmland. SQ and health status exert direct influences on human production, life, and ecological safety, and an understanding of SH status and its spatial distribution characteristics in farmland has vital practical significance for managing and protecting farmland SH, as well as ensuring China's food security and ecological civilization construction [11]. SH evaluation has gradually become an important tool to achieve this requirement [12], and research on the evaluation of SH based on functional soil management is essential [13,14].

Health serves as a necessary condition for the normal functioning of an organism or a part of an organism. As the medium for crop growth, soil sustains both physical and chemical activities on account of the long-term accumulation of organic matter and living organisms (such as insects, microorganisms, etc.), and is a life cycle in a dynamic equilibrium state [15,16]. Soil offers support to significant global ecosystem services, such as water purification, carbon sequestration, nutrient cycling, and providing habitat for biodiversity [17,18]. Soils contribute to general ecosystem services, defined as "services to society that ecosystems provide," which require collaboration among different disciplines [19]. The challenge for soil science is to explore ways in which healthy soils can contribute to improving a number of key ecosystem services that, in turn, can support SDGs [20]. It must be noted that SDGs and the goals of the EU Green Deal are not only determined by ecosystem services, but also by, for example, socioeconomic and political factors that are beyond the control of sciences studying crop growth. Thus, attention to the SDGs and EU Green Deal not only implies consideration of biomass production (SDG 2-zero hunger), but also other ecosystem services that are directly related to environmental quality, such as the quality of ground and surface water (SDG 6-clean water and sanitation), carbon sequestration and reduction of greenhouse gas emissions for climate mitigation (SDG 13-climate action), and biodiversity preservation (SDG 15-life on land). Thus, SH is the actual capacity of a particular soil to function, contributing to ecosystem services, while SQ is the inherent capacity of a particular soil to function, contributing to ecosystem services. Both these general definitions focus on soil contributions to ecosystem services that, in turn, promote the realization of SDGs and the goals of the EU Green Deal [9,21]. Bünemann et al. reviewed the differences between the widely used terms, SQ and SH, and concluded that SQ represents SH in most of the previous studies [22]. In addition, it must be noted that SQ includes inherent and dynamic qualities. Inherent SQ is related to the natural composition and properties of the soil (i.e., soil texture) and is significantly influenced by long-term natural factors and soil-forming processes [23]. In contrast, dynamic SQ is equivalent to SH, including soil properties that change on a human time scale owing to soil use and management, such as organic matter, bulk density, and aggregate stability [24]. It demonstrates that SH and SQ have a relationship of mutual conversion with their respective emphases. Leopold Aldo proposed for the first time that SH represents



the ability of soil to self-regulate and renew [25], while Trutmann et al. suggested that SH denotes the ability of soil to play the role of an important life system within the boundaries of ecosystems and land use [26]. With expanding research, the concept of SH was further improved, indicating that soil serves as a crucial ecosystem that supports the survival of plants, animals, and humans, and that SH refers to the ability of the soil to play this role. In 2016, the idea of SH status, defined as the continuous ability of soil as a dynamic living system to maintain its functions and represented by cleanliness and biodiversity, was introduced into the "Cultivated Land Quality Grade" (GB/T 33469-2016) in China. Subsequently, Chinese scholars have successively performed studies on land health [27], farmland health [28], and farmland health productivity [29], which have enriched our understanding of SH. The evolution of the concept of SH is a process in which people continue to deepen their understanding of soil and its interaction with agriculture, environment, ecology, and climate change; strengthen innovative soil science research and collaboration between soil science and related disciplines; actively call on the whole society to cherish and protect the non-renewable resource of soil, and promote sharing of soil science achievements and coordination of soil utilization and management policies; and encourage more stringent soil protection legislation to realize SDGs and the EU Green Deal.

SH can be summarized as the ability of soil to perform soil functions (SFs) and provide ecosystem services. SF is a series of soil processes that support the provision of ecosystem services [30], and soil stress serves as a significant factor affecting these processes [31]. In recent years, studies on SH/SQ evaluation gave involved SF, ecosystem services, and soil stress [32,33]. Since the beginning of this century, the Soil Management Assessment Framework (SMAF) proposed by Andrews et al. of the United States Department of Agriculture has been widely used worldwide [34] and has been continuously improved and developed to have a profound influence on the research on SQ [35]. Under the guidance of this theoretical framework, SH evaluation has adopted a general technical route of "setting evaluation goals—clarifying evaluation objects and SFs involved—selecting evaluation indicators and methods—outputting evaluation results". Soil has multiple functions and plays an important part in agricultural production and maintaining stability of terrestrial ecosystems. Similar to the role of human body function in human health evaluation, SF is significant in SH evaluation, and is a vital component of SH. The indicators involving soil properties, topography, and climatic conditions are selected to analyze and integrate the evaluation results of multiple SFs [36], using qualitative and quantitative models and digital mapping technology of soil to accurately represent a single SF, which lays the basis for SH evaluation. Germany, which paid attention to SF earlier, has provided a national-scale practice plan, with each state independently performing SF evaluations to serve the actual needs of local natural resource management within a certain framework, covering multiple aspects such as landscape, agriculture, environment, and soil protection [37]. Healthy and sustainably managed soils provide many benefits to people, nature, and climate. Besides, healthy soils are essential for delivering healthy food and other essential ecosystem services to humans, such as biomass production, purification of percolating water, avoidance of surface water pollution, reduction of greenhouse gas emissions, carbon capture for climate mitigation, and preservation of biodiversity [21]. Soil must be evaluated for multiple functions, rather than for productivity alone [38]. Currently, although there are no similar SF classification schemes between China and other countries, the main SFs related to SH/SQ are almost identical, namely, the functions closely related to agricultural and forestry production, such as primary productivity (PP), provision and cycling of nutrients (PCN), provision of functional and intrinsic biodiversity (PFIB), water purification and regulation (WPR), and carbon sequestration and regulation (CSR) [39–42]. Therefore, an integrated approach is needed and can be achieved by simulating the soil–water–atmosphere–plant (SWAP) system when considering the "functioning of soils" [9]. The application of simulation models of the SWAP system can integrate the values of the parameters, producing a single integrated value for biomass production. Many operational models are currently available [43,44], which use rooting depth and weather data, among

others. When the required hydraulic conductivity and moisture retention data are not available, these values can be estimated with pedotransfer functions using texture (as defined by the soil type), organic matter, and bulk density as the input data [45].

Driven by the development of soil science theory and upgrade of technical approaches, the methods for evaluating soil based on soil properties and processes have been significantly improved. This advancement lies in the integration of prior knowledge of soil basic sciences to establish the mapping relationship between various SF and soil properties and processes, and then measures the actual level of SF through hard data. For example, Thoumazeau et al. proposed an integrated indicator set-Biofunctool® to evaluate the impact of agricultural land management on SF, which consists of 12 fast and economical field indicators, including soil active organic carbon, soil basic respiration, earthworm activity, available nitrogen, and the infiltration rate and stability of soil aggregates, to assess three dynamic SFs involving carbon conversion, nutrient cycling, and structure maintenance [46,47]. Nowadays, although there are practical and feasible SH evaluation programs such as the Comprehensive Assessment of SH-The Cornell Framework [23], SH Card [48], etc., none clarify the target SF and only help to meet the needs of soil utilization and management at the field scale. In terms of the integration of multiple function evaluation results, Schulte and Bampa et al. conceptually described for the first time the cooperative tradeoffs of SFs of various types of land use [49,50], such as farmland, forest land, and grassland, on the basis of the theory of functional soil management and concluded that types of land use have different requirements for various SFs. This theory preliminarily explains the relationship between SF and SH, integrates the results of comprehensive SH evaluation to overcome the long-term lack of separation of SH evaluation indicators and SFs, and provides a new idea: the evaluation of SH should consider the demand of land-use type for SF, i.e., attention must be paid to the supply capacity of SF related to the type of land use. Regarding practical application, the Soil Navigator Farmland Management Decision Support System, recently developed by the EU Horizon2020-LANDMARK project, is based on a decision model that combines expert experience and data mining to evaluate a variety of SFs and provide farmers and farm consultants with targeted farmland management solutions on the basis of the users' input data embodying agricultural ecosystems, management, environment, and soil [51]. Owing to the increasing utilization of farmland and large input of foreign substances such as chemical fertilizers and pesticides, which are not only limited to SQ degradation, but also soil environment. As a result, the continuous restoration and maintenance of the optimal function of soil and improvement of SH are crucial with respect to agricultural production. The ultimate goal of soil multifunctionality management is to ensure food self-sufficiency and food security. Accordingly, the purpose of the current study was to develop an effective, scientific, and reasonable SH evaluation system based on the concept of functional soil management, to provide support for the sustainable use and management of soil resources, and to improve SH evaluation program in China. The specific purposes of the current study were as follows: (i) the establishment of an SH evaluation system based on the concept of functional soil management; (ii) the improvement of the previous score and gradation scheme of comprehensive index method, and the combination of the minimum limiting factor method with the weighted linear model; and (iii) a quantitative analysis of the obstacle factors affecting SH of farmland with the application of the obstacle degree model. The potential scientific contributions of the study include: (A) matching the SFs related to SH and establishing a new SH evaluation system; (B) choosing evaluation methods that are conducive to the actual management and operation of the new SH evaluation system; (C) highlighting the importance of limiting factors or risk/hazard factors; (D) improving the comprehensive index method with the adoption of a least limiting factor method of limiting factors, a weighted linear model of non-limiting factors, and a mathematical algorithm integrating the results of limiting factors and non-limiting factors; and (E) utilizing the food production stability index to verify the feasibility of the SH evaluation program. Yixing City, Jiangsu Province, China, is a commercial grain base in the Yangtze River Delta and large rice planting county serving

as an important granary in Jiangsu Province. However, in recent years, industries such as ceramics, textiles, and mining have caused heavy metal pollution of the agricultural land in this area. Considering Yixing City as an example, we conducted field tests to verify the new SH evaluation system (Figure 1) as well as provide a reference for systematic regional farmland SH research and promote food security, ecological protection, and sustainable agricultural development.

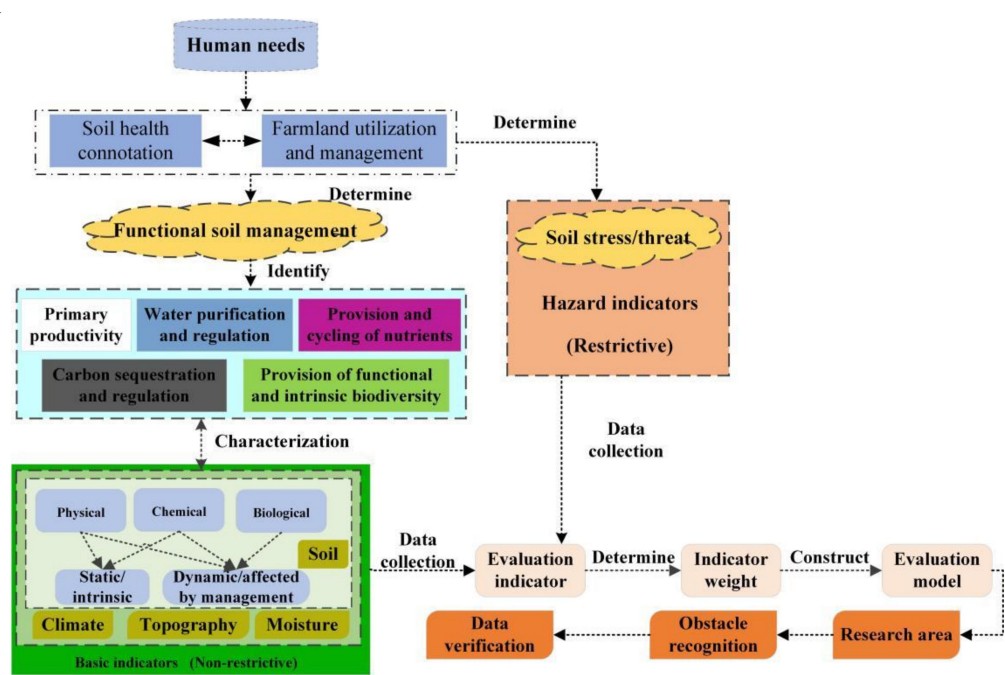

**Figure 1.** Research framework.

## 2. Materials and Methods

### 2.1. Study Area

Yixing City is located at the north latitude of 31°07′–31°37′ and east longitude of 119°31′–120°03′, with a complex topography and various types of landforms. The terrain is high in the south with low hills and low in the north with broad plains, with an average elevation of low mountains ranging from 150 to 240 m, and the hills generally bearing an elevation of about 50 m. Situated in the northern subtropical monsoon climate zone, Yixing City has a mild climate and four distinct seasons, with an annual average temperature of 15.7 °C, annual average frost-free period of more than 240 days, annual average precipitation of 1199.5 mm, and decreasing pattern of rainfall from south to north. The city has a population of 1.0833 million, with 14 towns and four streets. The farmland covers an area of 53,800 ha, accounting for 26.7% of the total land area. Four suborders of soil are found in the area of Yixing according to the Chinese Soil Taxonomy (Revised proposal), namely, Stagnic Anthrosols, Udic Argosols, Udic Cambosols, and Orthic Anthrosols, with Stagnic Anthrosols being the main soil type in the area [52]. According to the Soil History (Farmer Naming) of Yixing City, the soil in this region is divided into seven types (yellow brown soil, red loam, limestone soil, purple soil, paddy soil, fluvo-aquic soil, and swamp soil), 13 subtypes, 31 soil genera, and 73 soil species. The soil in the low mountain areas in the south is dominated by yellow brown soil and red soil; the northwest is dominated by white soil; the middle and east of Gehu Lake is covered with yellow clay; the middle and western polder is concentrated with gleyed paddy soil; and the eastern ditch district is dominated by lake white soil and fluvo-aquic soil. With a farming system of two crops a year, the summer grains include wheat, barley, broadbean, and pea, and autumn grains comprise rice, corn, potatoes, and beans, among which wheat and rice are the predominant summer and autumn grains, respectively. In recent years, industries such as ceramics,

textiles, and mining in Yixing City have put excess pressure on the environment, and the damage to the soil environment is becoming increasingly significant, causing serious soil heavy metal pollution in the agricultural land. Hence, with regard to food production and ecosystem services, the utilization of national soil resources must be based on soil and its core functions to guarantee healthy and sustainable soil.

*2.2. Data Sources*

The data on soil heavy metals pollution and some physical and chemical properties were principally derived from the "Technical Requirements for Geochemical Assessment of Land Quality (Trial)" (DD2008-06). A total of 884 points were established with a sampling density of about two points per square kilometer (Figure 2). Grain production data were obtained from the "Statistical Yearbook of Study Area" (2009–2018), and 115 soil samples were collected in 2015. With the sampling points centered on GPS positioning points, a pattern of "S" or checkboard was established based on the plot shape, radiating 50–100 m around, to ascertain the sample points, and a mixed soil sample comprising 1 kg of soil from the sample points was collected using the quarter method. After removing rocks and other debris, the soil samples were naturally air-dried and analyzed for soil pH, total organic carbon, CEC, and six heavy metals (As, Cd, Cu, Hg, Zn, and Pb). To examine the heavy metals contents in crops, 96 crop samples were collected, including 45 samples of rice and 51 samples of wheat. Following the principles of comprehensiveness, representativeness, objectivity, equilibrium, comparability, and accessibility, 110 soil sampling points were established based on a stratified sampling method in 2018 to examine soil respiration by analyzing the soil moisture content, soluble organic carbon, and aggregate stability. Different types of attributes of Yixing City were considered, including ground class, soil type, and land use status. The surface layer (0–20 cm) of the cultivated soils were sampled before soil fertilization following the autumn harvest of crops, using a handheld GPS for coordinate positioning. A Kriging interpolation value with less error was selected by cross-validation comparison of the inverse distance weight and Kriging value, and spatial distribution maps of each indicator were generated after test correction [53]. The soil earthworms were collected through freehand separation and the number of collected soil earthworms was recorded. The indicator data of soil intrinsic properties were obtained from the soil survey report of the study area and "Chinese Soil Series-Jiangsu Volume", and the basic geographic data were downloaded from the National Earth System Science Data Center, National Science & Technology Infrastructure of China (http://www.geodata.cn/, accessed on 16 November 2016). The elevation data of the study area were primarily obtained from the high-resolution remote sensing data of 2016 GDEMDEM 30M Resolution in the geospatial data cloud (http://www.gscloud.cn/, accessed on 16 October 2016). A total of 115,479 polygons of farmland in the map layer in the 2016 land use change survey database of Yixing City were selected as the evaluation unit.

*2.3. Overall Framework of SH Evaluation*

With the introduction of concepts such as SH, researchers are increasingly describing SH from the perspective that soil provides a variety of ecosystem services [54]. SF and its ecosystem service are a part of SH evaluation method principally used for land use and management and are the main topics of SH evaluation research in recent years [55]. It is believed that healthy farmland soil can produce healthy agricultural products, and presents properties such as non-pollution, better buffering and filtering, high elasticity, strong resistance, good recovery, and sustainability. In other words, healthy farmland soil allows the normal performance of five SFs of PP, PCN, PFIB, WPR, and CSR. In the process of soil utilization, excessive input of production factors such as pesticides and fertilizers and improper use of agricultural films may disturb the soil ecosystem, causing a series of SH problems, including soil erosion and soil pollution [56]. Thus, the restriction of one or more SFs may endanger SH (e.g., soil compaction, erosion, loss of biodiversity, loss of organic matter, salinization, pollution, and desertification) and have an impact on the reasonable

use and protection of SH. In the German Müncheberg SQ evaluation system, both the basic indicators (non-restrictive indicators) and hazard indicators (restrictive indicators) are taken into account, with the former being related to soil elements and soil structure, and the latter being associated with SF limitations. The final SQ score is calculated by multiplying the sum of basic indicators by the product of restrictive indicators [57]. If a restrictive indicator is over-constrained, the final score will be extremely low, and a highly restrictive indicator will have a decisive impact on the results of SQ evaluation. Therefore, in the present study, the German evaluation system was used as a reference for the overall framework of SH evaluation, and the evaluation indicators were divided into two categories, namely, basic indicators and hazard indicators, as shown in Figure 3. The basic indicators include indicators related to soil fertility, environment, health, and crop growth, and their selection depends on SFs. The nature of farmland is also a key driving factor in determining whether the soil can provide these multiple functions. Each function embodies two properties, namely, the relatively stable inherent property influenced by natural soil-forming factors and the dynamic property that changes because of human use and management [58]. The hazard indicators represent indicators that hinder the normal growth and development of crops or severely limit the performance of SFs, such as soil pollution and soil acidification. The basic indicators have an additive relationship, while the hazard indicators bear a multiplicative correction relationship. As the hazard indicators have different degrees of influence on the inherent and dynamic properties of soil, this difference cannot be reflected through the application of the same multiplier; hence, in the current study, two sets of multiplier factors were employed to modify the basic value of SH. The use of SF and stress as the intermediate process to evaluate SH can not only reflect the fundamental functions of the soil based on basic indicators and guide functional land management but can also demonstrate the influence of limiting factors through hazard indicators to achieve regional SH management and protection.

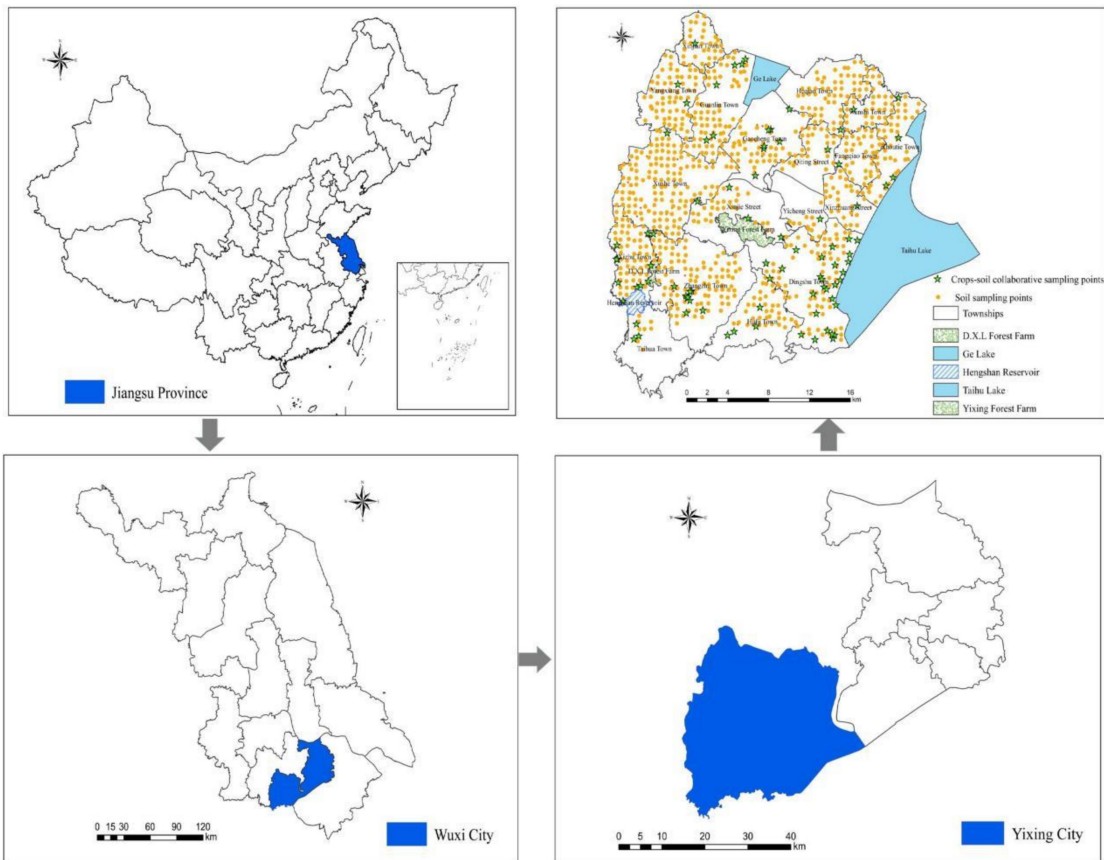

**Figure 2.** Location and distribution of the sampling points in Yixing City.

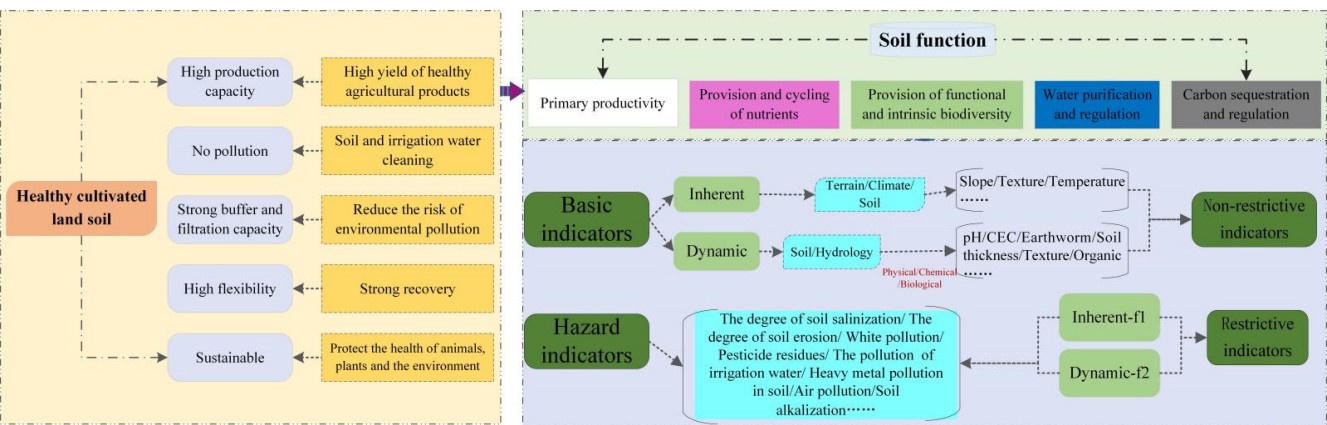

**Figure 3.** General framework for SH evaluation.

It must be noted that a combined physical movement of soil physical properties and climate and landform elements occurs. For instance, after reaching the ground surface, atmospheric precipitation undergoes infiltration, evaporation and re-precipitation through the soil pores to provide the necessary water for the growth of crops. Pollutants are buffered and filtered by chemical reactions, including redox, acid-base, and ion reactions, in the soil, to provide a good soil environment for the growth of crops. Furthermore, biological processes such as the mineralization of soil organic matter, humus formation, and carbon, nitrogen, and phosphorus cycles offer the required nutrient elements for crops growth; climatic conditions (sunlight and precipitation) provide the necessary light and heat for crops growth; and geomorphic elements (such as slope of the field) influence crops growth. Accordingly, the inherent attribute indicators can be selected from the conditions of climate, terrain, and soil, while the dynamic attribute indicators can be divided into physical, chemical, and biological indicators. The degree of influence of hazard indicators on inherent and dynamic attributes is represented by multipliers $f_1$ and $f_2$, respectively, the minimum multiplier in the hazard indicators is multiplied by the basic SH score, and the inherent and dynamic attribute scores are added to obtain SF index (SFI) of each SF. Subsequently, based on related research, weights are assigned to each function according to the relative importance of the SFs of the farmland to calculate the comprehensive SH index (SHI) of the study area.

### 2.4. Selection of Evaluation Indicator

Among many factors influencing SQ and SH, the established indicator system should follow the principles of stability, dominance, spatial variability, regionality, and systemicity, and select representative and influential indicators. In accordance with the overall framework of SH evaluation (Figure 3), a set of evaluation indicator systems was established (Table 1), among which the inherent attributes primarily included climate, topography, hydrology, and soil. Considering climate and topographic factors as significant geographical and environmental conditions that influence the formation and development of soil, annual accumulated temperature and annual precipitation indicators, which are closely related to agricultural production, were selected, and terrain indicators, including slope and landform type, were chosen to characterize the effect of hydrothermal conditions and topographic features on the SH of the study area. Owing to the difficulty in acquiring data, the soil moisture content indicator was selected as the hydrology condition to measure the soil water supply and drainage capacity. Moreover, based on previous studies [59,60], the basic evaluation indicators, such as soil layer thickness, soil body configuration, texture, soil aggregate stability, soil organic matter content, soil nutrient elements contents (nitrogen, phosphorus, and potassium), soil cation exchange capacity, soil pH, soil earthworm, and clay content were selected and classified into three categories of physical, chemical, and biological indicators.

**Table 1.** SH evaluation indicators system.

| Basic Indicator Non-Restrictive | Evaluation Indicators | | SF | | | | | Evaluation Indicators | | Basic Indicator Non-Restrictive |
|---|---|---|---|---|---|---|---|---|---|---|
| | | | PP | WPR | CSR | PFIB | PCN | | | |
| Inherent properties | Soil | Soil depth | | | | | | Bulk density | Physical indicators | Dynamic properties |
| | | Texture | | | | | | Clay content | | |
| | | Soil tillage depath | | | | | | Aggregate stability | | |
| | Topography | Soil configuration | | | | | | Organic content | Chemical indicators | |
| | | Slope | | | | | | Soil pH | | |
| | | Geomorphic type | | | | | | CEC | | |
| | Climate | Annual accumulated temperature | | | | | | Soil nutrient element | | |
| | | Annual precipitation | | | | | | Beneficial trace element | | |
| | | | | | | | | Soluble organic carbon | | |
| | | | | | | | | Soil respiration | Biological indicators | |
| | | | | | | | | Soil earthworm | | |
| | | | | | | | | Soil moisture content | Hydrology | |
| Hazard indicator (Restrictive) | The degree of soil salinization, depth of the barrier layer, degree of soil erosion, white pollution, pesticide residues, irrigation water pollution, heavy metals pollution, atmospheric pollution, degree of surface rock exposure, crop pollution, gravel content, soil acidification, soil compaction, flood disasters, and soil alkalization | | | | | | | | | |

Note: The box filled with color denotes that the attribute refers to an indicator for evaluating the corresponding SF. Green filling represents inherent attributes and yellow filling suggests dynamic attributes.

It must be noted that SH will inevitably suffer a certain degree of damage with the continuous development and utilization of soil resources by humans. In the process of soil utilization, the interaction of hazards caused by human activities and natural elements, including climate and terrain, can accelerate soil degradation and threaten the balance of soil ecosystems. In the past few years, China's one-sided pursuit of high yields of crops has resulted in the severe plunder of soil nutrients, excessive application of chemical fertilizers, and industrial and agricultural pollution, leading to continuous soil deterioration, especially soil compaction, acidification, and salinization, reduction in soil organic matter content, and excessive pesticide or antibiotic residues and heavy metals contents above the national standard. These problems can exert dramatic influences on the level of SH, causing a decrease in crop yields, decline in agricultural product quality, reduction in biodiversity, and even human health hazard. In view of the external attributes or environmental factors that limit the sustainability of SH, the indicators of SH hazards for farmland, including the degree of soil salinization, depth of the barrier layer, degree of soil erosion, white pollution, pesticide residues, irrigation water pollution, heavy metals pollution, atmospheric pollution, degree of surface rock exposure, crop pollution, gravel content, soil acidification, soil compaction, flood disasters, and soil alkalization, were selected in the present study [61–65].

The following are the evaluation indicators of each SF selected in the present study.

I.     PP

The function of PP is the production ability of soil to produce plant biomass for human use and to provide food, feed, fiber, and fuel within the boundaries of natural or managed ecosystems [66,67]. PP is an SF known to humans for a long time, and climatic conditions, topography, soil basic characteristics, and human activities all have a vital influence on the production function of soil. Climatic conditions control the growth and spatial distribution of crops and vegetation and determine the cropping system of plants; topographic features can affect the local climate and the difficulty or simplicity of agricultural production methods; and suitable soil moisture content enables improvement of the growth environment of crops and increases the level of food production. Soil organisms play an indispensable role in the productivity of farmland by providing a stable environment for the planting, growth, and maturation of crops, and the biological effects of soil generate the essential nutrients necessary for crops growth. Table 1 lists all the basic indicators applied in this study.

II.    WPR

WPR function is the ability of the soil to absorb, store, and transport water for later use, prevent long-term drought, floods, and soil erosion, and remove harmful compounds (such as volatile organic compounds and heavy metals) from water. Soil serves as a buffer zone and filter for pollutants, and the health risks of environmental pollution to humans, animals, and plants are reduced through adsorption, decomposition, and transformation [68]. Soil texture can retain soil moisture and block the movement of pollutants to a certain extent. Soil with higher organic matter and clay content presents better buffering and stronger capacity to preserve soil water and fertilizer [59]. The soil aggregate stability can determine the stability of soil structure and ability to resist erosion. A soil system with good aggregate stability is more flexible and plays an important role in maintaining the soil structure, preventing soil compaction, and resisting drought, runoff, and erosion risks, and presents better permeability and water storage capacity. The soil aggregate stability is usually represented by mean weight diameter (MWD), and higher MWD denotes increased aggregate stability [69]. Soil pH has an impact on the ability of soil to adsorb heavy metals and other pollutants [35], and soil moisture content affects infiltration and water absorption capacity of soil. In the present study, indicators that reflect the soil water retention and purification capacity, such as soil layer thickness, soil bulk density, soil texture, clay content, soil moisture content, tillage layer thickness, aggregate stability, organic matter content, soil pH, and CEC, were selected.

III.    CSR

The function of CSR is the ability of the soil to store carbon in an unstable form with the purpose of slowing down the increase in the concentration of carbon dioxide in the atmosphere, which has a decisive effect on climate regulation. The amount of soil carbon sequestration is principally reflected in the soil organic matter content. Soil layer thickness and soil texture are crucial factors that control soil carbon content and stability. When compared with sandy soil, soil with clay and silt can absorb more organic carbon owing to the ability of clay and silt to adsorb ions. Soil bulk density and pH are associated with soil carbon fixation rate. Besides, climatic conditions also have a significant impact on organic carbon sequestration, with cool and humid conditions causing higher organic carbon sequestration, and warm and dry conditions resulting in lower organic carbon sequestration [70]. Hence, during an evaluation of CSR function, the prime indicators that must be taken into account include annual accumulated temperature, annual precipitation, soil texture, soil depth, clay content, organic matter content, soil bulk density, soil pH, and so forth.

IV.    PFIB

The PFIB function includes numerous soil biological processes. Soil can provide habitats for animals, plants, and microorganisms, supporting their life activities and protecting biodiversity [71]. These soil organisms interact with the ecosystem and constitute a part of the natural resource of soil [72]. Soil organisms are affected by local hydrothermal and climatic conditions, with rich diversity noted under better soil hydrothermal condition. Furthermore, soil texture also affects soil biodiversity, and soil with a higher degree of sandification has lower fertility, poor biological environment, and lower biomass, whereas soil with high organic matter and soluble organic carbon content is more conducive to biological survival [73]. Thus, an evaluation of the PFIB function should mainly consider indicators such as annual accumulated temperature, annual precipitation, soil texture, organic matter content, soil moisture content, soluble organic carbon, soil pH, soil respiration, and soil earthworms.

V.    PCN

The function of PCN indicates the ability of soil to absorb and maintain nutrients, produce and retain the nutrients absorbed by crops, promote soil biochemical processes, and provide and retain nutrients [74]. Organic matter is the main source of soil nutrients, while soil layer thickness and soil pH affect the conversion of soil nutrients. The CEC of soil is responsible for the mobility of nutrients and their availability to plants, whereas soil texture has an impact on soil aeration, hydrothermal conditions, and the conversion of nutrients. Hence, for the assessment of this function, indicators such as soil texture, soil layer thickness, organic matter content, soil nutrient elements, CEC, and soil pH, were selected.

*2.5. Comprehensive Evaluation Method*

2.5.1. Determination of Indicator

Weight enables measuring the relative importance of each evaluation indicator and has a direct influence on the accuracy of the evaluation result. Some indicators of SH evaluation have dynamic attributes, and the contribution of each dynamic indicator to SFs varies. To eliminate the influence of different dimensions of the dynamic attribute evaluation indicator on the measurement and observation of the degree of change in the original data, the coefficient of variation method was applied to determine the objective weight of the dynamic indicator:

$$V_i = \frac{\sigma_i}{\overline{x}_i}(i = 1, 2, \ldots, n) \tag{1}$$

where $V_i$ is the variable coefficient of the *i*-th indicator, also known as the coefficient of standard deviation, $\sigma_i$ is the standard deviation of the *i*-th indicator, and $\overline{x}_i$ is the average number of the *i*-th indicator.

The weight of each dynamic attribute indicator can be given as follows:

$$W_i = \frac{V_i}{\sum\limits_{i=1}^{n} V_i} \tag{2}$$

An expert group comprising experts in land management, soil, and land information, among others was formed, and Expert Choice software was used to compare and determine the relative importance of each indicator and the five major SFs by employing an analytic hierarchy process and expert scoring to obtain the average value as the corresponding intrinsic attribute indicator weight after the indicator weights of all experts at all levels passed the consistency test. Based on the provision matrix of SF proposed by Coyle et al. [75], the weight of each SF in the SH evaluation of farmland was determined in this study using the soil utilization method, with the weight of PP, WPR, CSR, PFIB, and PCN reaching 0.6912, 0.0346, 0.0242, 0.0242, and 0.2258, respectively. With regard to qualitative indicators, each basic indicator was classified and determined through literature analysis, expert experience, etc. [76]. Table 2 provides detailed information on the grade scores of the basic indicators of SH evaluation.

2.5.2. Multipliers of Restrictive Factors

To highlight the significance of hazard indicators, soil without hazard risk or evident threat on SF was set as the benchmark with multiplier factor of 1. It must be noted that some differences exist in the degree of influence of various hazard indicators on the inherent and dynamic properties of soil. For instance, indicators such as soil heavy metal pollution, white pollution, and pesticide residues mainly affect soil dynamic properties; soil barriers and soil erosion have a certain impact on both inherent and dynamic attributes; and soil salinization factors have a relatively significant influence on dynamic attributes and moderately little effect on inherent attributes. In the presence of multiple serious (active) hazard factors, the smallest multiplier in the hazard indicators was selected as the multiplier factor. In accordance with the expert consultation method and related literature [77], the corresponding multiplier factors were separately designed based on the attributes (intrinsic and dynamic) of the indicators. As spatial scale has a greater impact on the setting of multiplier classification of hazard indicators, this study only considered the impact of hazard indicators of SH on the study area, and two multipliers were established according to the relative influence of soil inherent and dynamic properties, focusing on the comparative investigation and analysis of SH of the study area (Table 3).

**Table 2.** Grading reference for basic indicators of SH evaluation.

| Basic Indicator | The Gradation and Assignment of Basic Indicator (Non-Restrictive) | | | | | |
|---|---|---|---|---|---|---|
| | **100** | **90** | **80** | **70** | **60** | **40** |
| Slope (°) | <2 | [2, 6) | [6, 15) | | [15, 25) | ≥25 |
| Geomorphic type | Plain | | Hills | | Plateau | Mountain land |
| Annual precipitation | ≥1280 | (1250, 1280] | | <1250 | | |
| Annual accumulated temperature (≥10°) | ≥5400 | (5380, 5400] | (5360, 5380] | <5360 | | |
| Texture | Loam | Clay | | Sand | | Gravelly soil |
| Soil depth (cm) | ≥150 | [100, 150) | | [60, 100) | [30, 60) | <30 |
| Soil moisture content (%) | Loam [15,25), Clay [25,30) | | Loam [10,15), Clay [20,25) | | Loam <10 or ≥25, Clay <20 or ≥30 | |
| Soil pH | [6.0, 7.9) | [5.5, 6.0) or [7.9, 8.5) | [5.0, 5.5) or [8.5, 9.0) | | [4.5, 5.0) | <4.5 or ≥9.0 |
| Clay content (%) | >35 | (25, 35] | | (15, 25] | (10, 15] | ≤10 |
| Soil tillage depth (cm) | >20 | (15, 20] | | (10, 15] | | ≤10 |
| Aggregate stability | Water stable aggregate > 45% | | Water stable aggregate (30%,45%] | | Water stable aggregate ≤ 30% | |
| Organic content (g/kg) | ≥40 | [30, 40) | [20, 30) | [10, 20) | [6, 10) | <6 |
| Bulk density (g/cm³) | [1, 1.25) | [1.25, 1.35) | | [1.35, 1.45) | [1.45, 1.55) | ≥1.55 or <1 |
| Soil configuration | All loam, loam/clay/loam | Loam/sticky/sticky, loam/sand/soil, sand/sticky/sticky | | Sand/sticky/sand | Loam/sand/sand, Sticky/sand/sand | All sand, all gravel |
| CEC (cmol/kg) | >20 | [15, 20) | | [10,15) | | ≤10 |
| Soil nutrient element | $C_{\text{Comprehensive Score of Ntrients}} \geq 90$ | $90 > C_{\text{CSN}} \geq 75$ | | $75 > C_{\text{CSN}} \geq 60$ | $60 > C_{\text{CSN}} \geq 45$ | $C_{\text{CSN}} < 45$ |
| Beneficial trace element | $C_{\text{Beneficial}} \geq 90$ | $90 > C_{\text{B}} \geq 75$ | | $90 > C_{\text{B}} \geq 75$ | $90 > C_{\text{B}} \geq 75$ | $90 > C_{\text{B}} \geq 75$ |
| Soluble organic carbon (mg/kg) | ≥100 | [80, 100) | [60, 80) | [40, 60) | [20, 40) | <20 |
| Soil respiration (μgCO₂/(g·h)) | >1 | (0.8, 1] | (0.4, 0.8] | (0.2, 0.4] | (0.1, 0.2] | ≤0.1 |
| Soil earthworm (quantity/m³) | >20 | (15, 20] | (10, 15] | | (5, 10] | ≤5 |

Table 3. Grading reference for hazard indicators of SH evaluation.

| Hazard (Restrictive) Indicator | Grading Standard | Description of Hazard Grade | Multiplier | |
|---|---|---|---|---|
| | | | f1 | f2 |
| Degree of salinization | No risk of saline-alkaline soil or salinization | No salinization | 1 | 1 |
| | Soluble salts up to 0.1% | Mild | 0.9 | 0.8 |
| | Soluble salts up to 0.3% | Medium | 0.7 | 0.5 |
| | Soluble salts up to 0.5% | Severe | 0.4 | 0.2 |
| The depth of barrier layer from surface | >90 cm | No significant effect | 1 | 1 |
| | 60–90 cm | Low risk | 0.9 | 0.8 |
| | 30–60 cm | Medium | 0.7 | 0.6 |
| | <30 cm | High | 0.5 | 0.3 |
| Degree of soil erosion | The profile of three layers of A, B and C maintains complete | No significant erosion | 1 | 1 |
| | More than half of thickness of A layer is preserved, while the layer of B and C maintain complete | Mild | 0.9 | 0.9 |
| | More than one third of thickness of A layer is preserved, while the layer of B and C maintain complete | Medium | 0.7 | 0.7 |
| | Nothing of layer A is left, layer B begins to be exposed to erosion, and layer C maintain its completeness | Serious | 0.4 | 0.4 |
| | The layer of A and B is entirely denuded, while the layer C is exposed to erosion | Extremely | 0 | 0 |
| White pollution | The residual amount of mulch of agricultural film < 10 kg/hm$^2$ | The small amount of mulch residues of agricultural film | 1 | 1 |
| | 10–60 kg/hm$^2$ | The medium amount of | | 0.8 |
| | >60 kg/hm$^2$ | The large amount of | | 0.6 |
| Pesticide residues | No pesticide residues | No pesticide residues | 1 | 1 |
| | <0.3 mg/kg | Low | | 0.8 |
| | 0.3–1 mg/kg | Medium | | 0.6 |
| | >1 mg/kg | High | | 0.3 |
| Pollution of irrigated water | The comprehensive pollution index of water quality ≤ 0.5 | Clean | 1 | 1 |
| | 0.5–1.0 | Roughly clean | | 0.7 |
| | ≥1.0 | Polluted | | 0.3 |
| Heavy metal pollution | The comprehensive pollution index $p \leq 0.7$ | No pollution, clean soil | 1 | 1 |
| | $0.7 < p \leq 1.0$ | Mild pollution, roughly clean soil | | 0.9 |
| | $1.0 < p \leq 2.0$ | Mild | | 0.7 |
| | $2.0 < p \leq 3.0$ | Medium | | 0.5 |
| | $p > 3.0$ | Heavy | | 0.3 |
| Atmospheric pollution | Cd ≤ 3 and Hg ≤ 0.5 (mg m$^{-2}$ a$^{-1}$) | Clean | 1 | 1 |
| | Cd > 3 or Hg > 0.5 (mg m$^{-2}$ a$^{-1}$) | Not clean | | 0.3 |
| Crop pollution | Inferior to the standard of green food | Healthy | 1 | 1 |
| | Between the standards of green food and pollution-free food | Sub-healthy | 0.8 | 0.9 |
| | Superior to the pollution-free food | Unhealthy | 0.5 | 0.6 |

**Table 3.** *Cont.*

| Hazard (Restrictive) Indicator | Grading Standard | Description of Hazard Grade | Multiplier f1 | f2 |
|---|---|---|---|---|
| Degree of exposed surface rock | Rock exposure < 2% | No influence on cultivation | 1 | 1 |
| | Rock exposure 2–10%, the space between outcrops > 35 m | Having an influence on cultivation | 0.8 | 0.9 |
| | Rock exposure 10–25%, the space between outcrops 10–35 m | Having an influence on mechanized cultivation | 0.5 | 0.6 |
| | Rock exposure ≥ 25%, the space between outcrops 3.5–10 m | Having an influence on small-scale mechanized cultivation | 0.3 | 0.4 |
| Gravel content | The content of gravel > 2 mm in soil particle composition < 1% | Non-gravelly soil | 1 | 1 |
| | 1–30% | Gravelly soil | 0.9 | 0.7 |
| | 30–50% | Mild gravelly soil | 0.7 | 0.5 |
| | 50–70% | Medium gravelly soil | 0.5 | 0.3 |
| | >70% | Major gravelly soil | 0.3 | 0.1 |
| Soil acidification | Ferralsols, with low soil pH, is insensitive to acid | Low acidification rate | 1 | 1 |
| | Luvisols, with the soil pH of subacidity or neutrality, is sensitive to acid | Relatively low | | 0.9 |
| | Semi-Luvisols, with the soil pH of neutrality or alkalescence, is highly sensitive to acid | Relatively high | | 0.7 |
| | Calcareous, with soil pH > 7, is generally insensitive to acid | Generally high | | 0.5 |
| | Others, with soil pH varying greatly, is sensitive to acid | Extremely high | | 0.3 |
| Soil compaction | Bulk density 1.00–1.10 (g/cm$^3$) | Low surface soil compaction, directly used as cultivated soil | 1 | 1 |
| | 1.10–1.35 (g/cm$^3$) | Relatively low surface soil compaction, used as cultivated soil | 0.9 | 0.9 |
| | 1.35–1.50 (g/cm$^3$) | Slightly high soil compaction, used as cultivated soil after improvement | 0.8 | 0.7 |
| | 1.50–1.60 (g/cm$^3$) | Relatively high soil compaction, used as cultivated soil after improvement in 1–2 year | 0.7 | 0.5 |
| | ≥1.60 (g/cm$^3$) | High soil compaction, the relatively high time and economic cost of improvement | 0.6 | 0.3 |
| Flood disaster | The flood control standard of farmland ≥ 10 years | No influence on crop growth | 1 | 1 |
| | 5–10 years | Slight | 0.9 | 0.9 |
| | 3–5 years | Medium | 0.7 | 0.7 |
| | <3 years | Great | 0.4 | 0.4 |
| Soil alkalization | The degree of alkalization: <5% | No alkalization | 1 | 1 |
| | 5–10% | Slight | | 0.8 |
| | 10–15% | Medium | | 0.6 |
| | 15–20% | High | | 0.4 |
| | >20% | Alkali soil | | 0.2 |

### 2.5.3. SHI Evaluation Model

The previous scoring and grading scheme of the comprehensive index method was improved in the present study, and the minimum limiting factor method was combined with the weighted linear model to acquire different SFI and SHI:

$$SFI = I_i * w_i * f_1 + D_j * w_j * f_2 \tag{3}$$

$$SHI = \sum_{r=1}^{5} SF_r * w_r \tag{4}$$

where $I_i$ is the score of the inherent attribute indicator, $D_j$ is the score of dynamic attribute indicator, $f_1$ and $f_2$ are the multiplier factors of the corresponding hazard indicator, respectively, and $w$ represents the weight of the indicator and SF. Through the natural breakpoint method, SF and SH were divided into three levels based on *SFI* (good, medium, and poor) and *SHI* (healthy, sub-healthy, and unhealthy).

### 2.5.4. Calculation of Food Production Stability Index

Food production stability index reveals the fluctuation and stability of grain yield [78] and is closely related to the five major SFs of cultivated SH and can measure the safety of the cultivated soil system structure. A higher food production stability coefficient indicates smaller change and fluctuation of grain output and healthier cultivated soil system. A village's food production stability index S can be calculated as follows:

$$S = 1 - \left[ n \sum_{i=1}^{n} \sqrt{(D_i - D)^2} \right] / D \tag{5}$$

where $D_i$ is the grain yield per unit of the i-th year, $D$ is the average grain yield per unit, and $n$ is the number of selected years; in the present study, 10 years were used as a stable period of measurement.

### 2.5.5. SH Obstacles Diagnosis Model

The obstacle degree model was introduced in the present study, and the obstacle factors of SH of the study area were quantified using factor contribution rate ($V_{ij}$), indicator deviation degree ($B_{ij}$), and obstacle degree ($M_{ij}$). $V_{ij}$ represents the degree to which the single indicator $j$ in the SF $i$ has an impact on the evaluation system [79], namely, the weight of the evaluation indicator $j$; $B_{ij}$ refers to the gap between the single indicator $j$ in the SF $i$ and its ideal value, which suggests the difference between the membership degree of the evaluation index ($A_{ij}$) and 1; and $M_{ij}$ indicates the degree of obstacle effect of the single indicator $j$ in the SF $i$ on SH. The indicator barriers could be divided into four categories, barrier-free (0), mild barriers (0, 10%), moderate barriers (10%, 20%], and severe barriers (20%, 100%), and can be calculated as follows:

$$M_{ij} = (1 - A_{ij}) * V_{ij} / \sum_{j=1}^{n} (1 - A_{ij}) * V_{ij} * 100\% \tag{6}$$

## 3. Results

### 3.1. SF and SH Evaluation

3.1.1. Comparative Analysis of SF and SH

Based on the proposed evaluation indicator system and evaluation model, the SFI and SH of the study area in Yixing City were calculated using ArcGIS 10.2 and classified according to Jenks Natural Breaks Classification (Table 4). The results of the spatial distribution and quantity structure of SF and SH are illustrated in Figure 4.

**Table 4.** Gradation of SF and SH in Yixing City.

| Gradation of SF | Good | Medium | Poor | Minimum Value | Maximum Value | Average Value | Mid-Value | Standard Deviation |
|---|---|---|---|---|---|---|---|---|
| I: PP | 58.98–98.03 | 46.51–58.98 | 32.66–46.52 | 32.66 | 98.03 | 53.55 | 50.11 | 10.46 |
| II: WPR | 58.54–96.99 | 45.59–58.54 | 32.45–45.59 | 32.45 | 96.99 | 56.46 | 50.09 | 10.61 |
| III: CSR | 63.42–100 | 50.01–63.42 | 39.27–50.01 | 39.27 | 100.00 | 59.86 | 55.46 | 10.64 |
| IV: PFIB | 55.31–92.78 | 43.71–55.31 | 31.01–43.71 | 31.01 | 92.78 | 46.92 | 45.50 | 10.01 |
| V: PCN | 54.74–95.93 | 42.99–54.74 | 30.25–42.99 | 30.25 | 95.93 | 48.77 | 45.74 | 9.95 |
| **Gradation of SH** | **Healthy** | **Sub-healthy** | **Unhealthy** | **Minimum Value** | **Maximum Value** | **Average Value** | **Mid-Value** | **Standard Deviation** |
| SH | 57.96–94.83 | 45.70–57.96 | 32.34–45.70 | 32.34 | 94.83 | 50.74 | 49.21 | 10.32 |

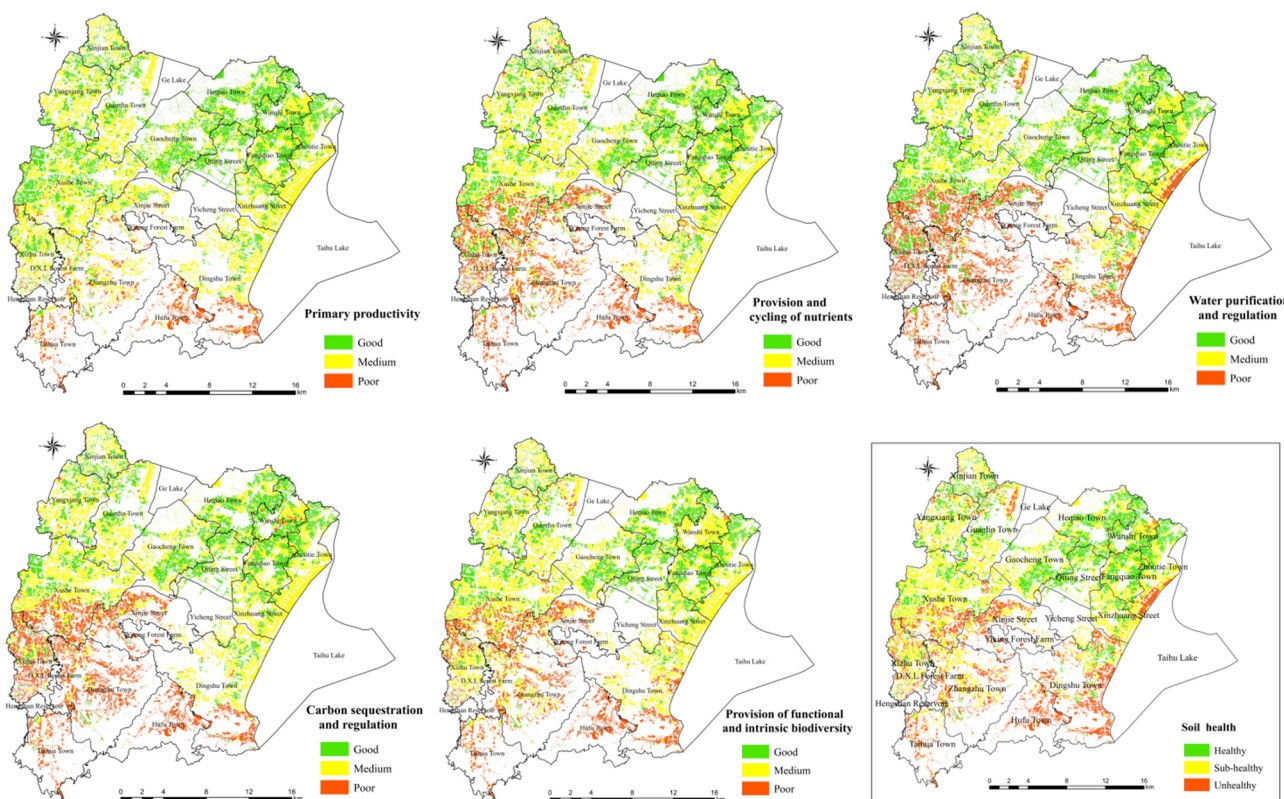

**Figure 4.** Spatial distribution map of SF and SH in Yixing City.

The results of the spatial distribution and quantity structure of SF and SH in Yixing City revealed that the healthy cultivated soil and optimally functional soil were predominant in the northeastern and mid-western regions of Yixing City, whereas poor and unhealthy soils were prevalent in the southern region of Yixing City. Besides, the function of WPR was poor in some soils along the periphery of Lake Taihu [80], where the soil was noted to be unhealthy.

From the perspective of quantitative structure, soils with various good functions were noted to cover an area of 21,345.49 ha (39.68%), 21,155.48 ha (39.04%), 16,963.22 ha (31.54%), 14,883.14 ha (27.67%), and 18,461.08 ha (34.32%); soils with poor functions covered 4076.76 ha (7.58%), 18,610.51 ha (34.6%), 11,796.32 ha (21.93%), 9526.48 ha (17.71%), and 10761.08 ha (20.01%); and the rest of the soils were medium-function soils. The quantitative structure of soil PP function, CSR function, PFIB function, and PCN function were classified as middle level, whereas WPR function reached a good level. The average SH was 50.74, with sub-healthy soil covering an area of 26,787.17 ha (49.80%), followed by healthy soil (15759.82 ha; 29.30%) and unhealthy soil (11,240.60 ha; 20.90%).

### 3.1.2. Spatial Distribution Characteristics and Pattern of SH

As shown in Figure 5, the SH levels in Yixing City varied. Healthy soils were noted to be predominant in Heqiao Town, Gaocheng Town, Zhoutie Town, and Wanshi Town in the northeast of the research area, accounting for 15.18%, 11.11%, 8.93%, and 8.35% of the city's healthy soil area, respectively. These villages and towns have not been affected by soil erosion and heavy metal pollution, and the farmland soils were dominated by yellow soil, gray soil, etc. The parent material of these soils was loess-like alluvium or lake sediment, which was one of the earlier soil types for planting and human utilization and exhibited characteristics such as deep and barrier-free profile, sticky texture, good fertility, high soil organic matter and nutrient content, and superior soil microbial activity. Sub-healthy soils were found to be principally distributed in Xushe Town in the mid-west and Guanlin Town, Gaocheng Town, and Zhoutie Town in the northeast of the study area, accounting for 21.68%, 7.44%, 7.23%, and 7.04% of the city's sub-healthy soil area, respectively. The soil types in these towns included gleyed paddy soil, fluvo-aquic soil, black beach soil, etc., with relatively flat terrain, loamy soil texture, higher soil microbial content, and soil fertility. Unhealthy soil was detected in Xushe Town in the mid-west, Dingshu Town in the southeast, Zhangzhu Town, and Hufu Town in the mid-south, and Xinjie Street in the central part of the research area, accounting for 25.22%, 13.80%, 8.89%, 10.94%, and 12.24% of the city's unhealthy soil area, respectively. A part of the farmland was observed to be exposed to heavy metal pollution, especially Cd pollution. In addition, the southern region with undulating terrain and many hills was noted to be dominated by yellow brown soil and red soil, which presented characteristics such as low soil pH, sandy soil texture, poor living environment for soil microorganisms, weak water storage capacity of soil, low organic matter content, and nearly low content of potassium and phosphorus. Besides, the southern soil also exhibited mild and moderate erosion, which can exert serious effects on the growth of crops [81].

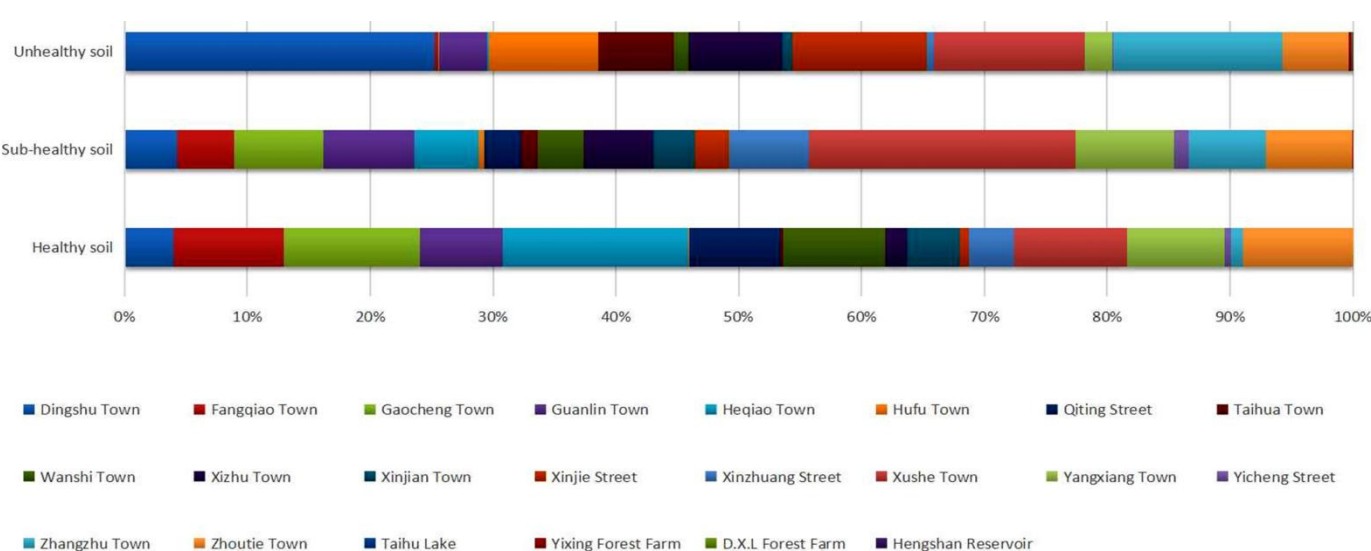

**Figure 5.** Comparison map of township distribution with different SH levels in Yixing City.

### 3.2. Diagnosis and Analysis of SH Obstacles

The obstacle degree model was employed to diagnose the obstacle factors of the basic indicators of SH evaluation in the study area. As some indicators, such as slope, topography, annual accumulated temperature, annual precipitation, soil layer thickness, soil bulk density, soil texture, clay content, cultivated layer thickness, aggregate stability, and soil body configuration, could not be improved, the diagnosis of obstacle degree was applied to nine evaluation indicators, namely, soil organic matter content, soil pH, CEC, soil nutrient elements, soil beneficial trace elements, soluble organic carbon, soil respiration,

soil earthworms, and soil water content (Figure 6). Regarding spatial distribution, the distribution of SH evaluation indicators significantly varied, especially CEC, nutrient elements, and soluble carbon.

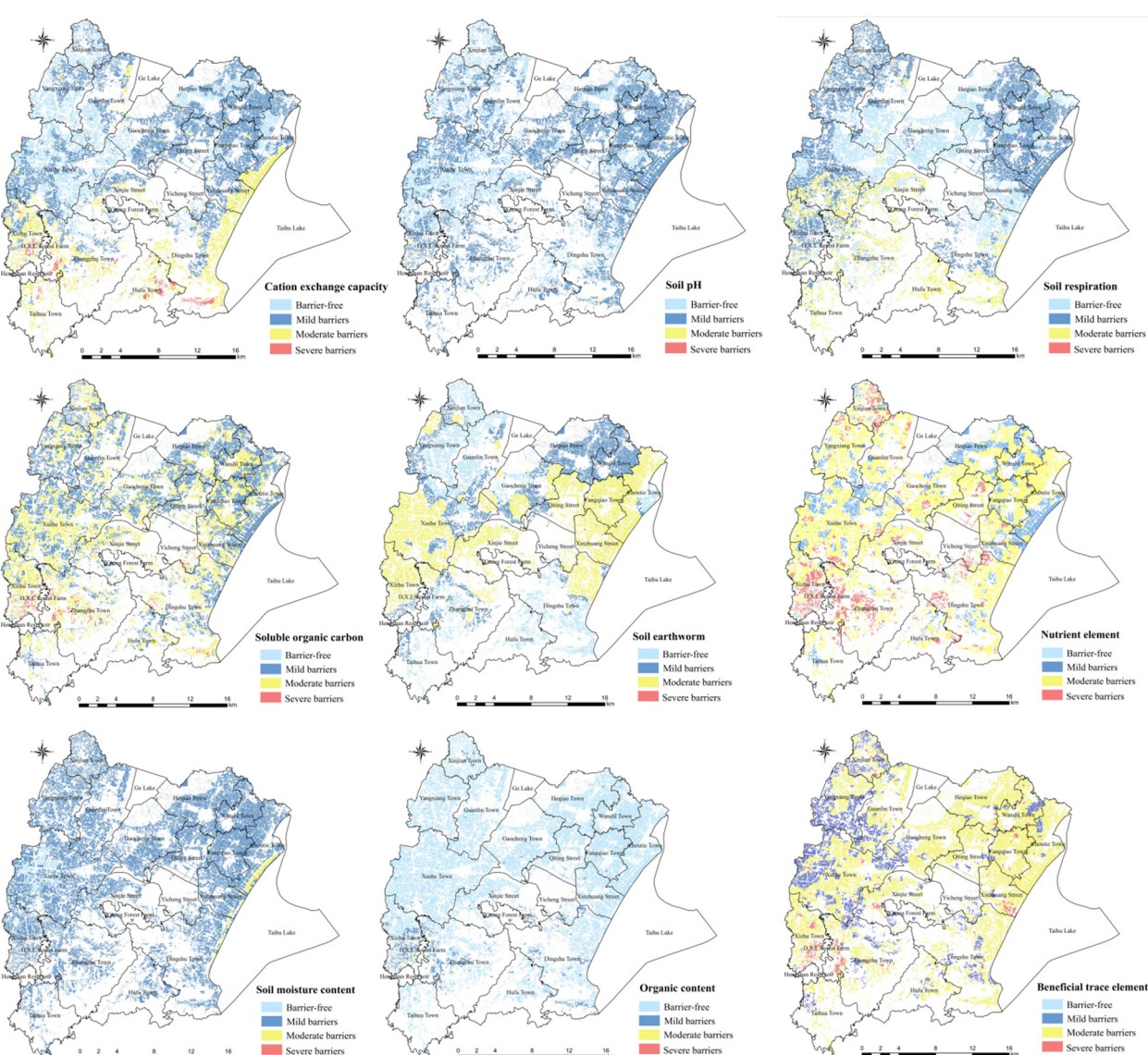

**Figure 6.** Spatial distribution of obstacle degree of some SH evaluation indicators in Yixing City.

Table 5 shows the obstacle degree of farmland SH evaluation indicators. The obstacle degree of soil organic matter was small, and 3296.78 ha (6.13%) of farmland, mainly distributed in Xizhu Town and Daxianling Forest Farm in the southwest of the study area, was noted to require protective method for cultivation, including increase in the use of organic manure and straws return in the future [82]. The soil pH of 35,952.78 ha (66.84%) of farmland scattered throughout the study area presented mild obstacles, and lime and sulfuric acid powder can be utilized to improve SH. CEC exhibited sporadic severe obstacles in Hufu Town, Dingshu Town, and Zhangzhu Town in the south of the study area, covering an area of 1185.83 ha (2.20%). The soil nutrient elements showed maximum moderate obstacles scattered throughout the study area, covering 36,504.17 ha (67.87%), and a combination of organic fertilizer and green manure can be used to promote soil fertility [83]. Soluble organic carbon exhibited mild and moderate obstacles, covering an area of 27,393.44 ha (50.93%) and 24,988.17 ha (46.46%), respectively, and soil amendments (such as biomass carbon) can be utilized to improve soil adsorption performance and

restore soil vitality [84]. Moreover, deep ploughing and scarification can be applied to improve the physical properties of soil, increase soil aeration, water permeability, and fertilizer retention capabilities, and reduce the input and utilization of production factors such as chemical fertilizers and pesticides as much as possible to decrease the intensity of soil utilization [85].

**Table 5.** Analysis of obstacle degree of SH evaluation indicators in Yixing City.

| Indicators | No Obstacles | Ratio% | Mild Obstacle | Ratio% | Medium Obstacle | Ratio% | Serious Obstacle | Ratio% |
|---|---|---|---|---|---|---|---|---|
| Organic content | 50,321.28 | 93.56 | 3296.78 | 6.13 | 168.63 | 0.31 | 0.90 | 0.00 |
| pH | 17,552.06 | 32.63 | 35,952.78 | 66.84 | 282.75 | 0.53 | 0.00 | 0.00 |
| CEC | 13,953.99 | 25.94 | 27,779.17 | 51.65 | 10,868.59 | 20.21 | 1185.83 | 2.20 |
| Nutrient element | 0.00 | 0.00 | 10,032.44 | 18.65 | 36,504.17 | 67.87 | 7250.98 | 13.48 |
| Beneficial trace element | 0.00 | 0.00 | 11,511.50 | 21.40 | 40,542.74 | 75.38 | 1733.35 | 3.22 |
| Soil moisture content | 2213.28 | 4.11 | 50,671.92 | 94.21 | 902.39 | 1.68 | 0.00 | 0.00 |
| Soluble organic carbon | 286.80 | 0.53 | 27,393.44 | 50.93 | 24,988.17 | 46.46 | 1119.19 | 2.08 |
| Soil respiration | 17,060.90 | 31.72 | 27,200.22 | 50.57 | 9526.48 | 17.71 | 0.00 | 0.00 |
| Soil earthworm | 12,609.78 | 23.44 | 9084.38 | 16.89 | 32,093.43 | 59.67 | 0.00 | 0.00 |

### 3.3. Relationship between Food Production Stability Index and SHI

In this study, an SH evaluation system was developed from the perspective of functional soil management, and the hazard indicators that limit SF were taken into account to reveal the differences between the regions. As soil productivity is the most basic and important function of cultivated soil, with healthy soil system exhibiting more stable soil productivity [86,87], we used the crop performance validation method to check the SH evaluation method [76]. The correlation between food production stability index and average SHI of each village in the study area was analyzed (Figure 7). The food production stability index of each village in Yixing City ranged from 0.09 to 0.22 in 2009–2018, and the average SHI range was 40.66–78.23 with a total of 316 data sets. An obvious correlation was noted between the average values of food production stability index and SHI with a relative $R^2$ value of 0.4607, which indicated that the evaluation system established in this study can reflect the health status of the local soil system and is reasonable. It must be noted that the present study only verified the stability of the most basic production functions of the soil and achieved a good fitting effect. In future, more relevant data must be obtained to verify other SFs, and human management and other factors should be appropriately considered.

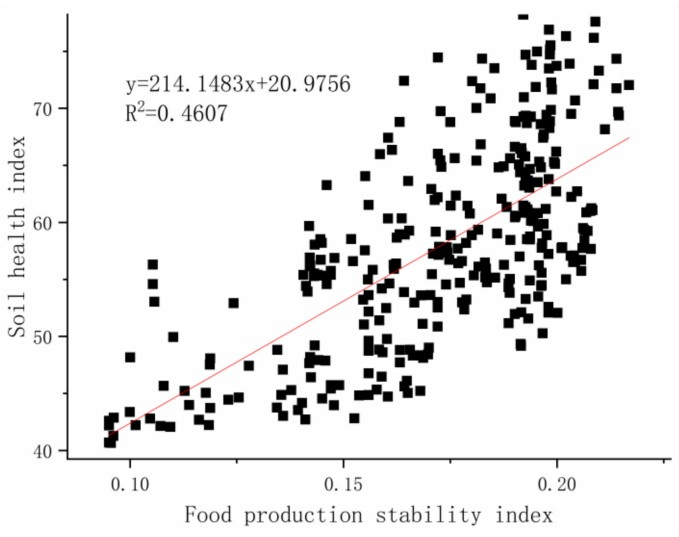

**Figure 7.** Correlation between food production stability index and SHI.

## 4. Discussion

Soil science integrates biology, hydrology, ecology, environmental science, geochemistry, geology, atmospheric science, etc., to provide a systematic explanation for the interaction among various layers of the earth and comprehensively manage SH. To derive meaningful contributions to general ecosystem services, collaboration among soil scientists, agronomists, hydrologists, climatologists, ecologists, social scientists, and economists in interdisciplinary research is desirable. As an ecosystem full of life, soil has different health conditions under diverse spatial scopes, times, and management methods, similar to human beings [88,89]. Therefore, SH not only requires macro-level control, but also micro-level implementation. Various scales (macro, meso, and micro) of SH evaluation meet the demands of decision-making and management activities of SH protection at different levels. As different scales solve different problems and soil is the resultant of the combined effect of soil-forming factors at diverse scales, the evaluation criteria for SH are varied at different spatial scales [90,91], with significant differences in the evaluation methods and effects. Hence, it is ideal to select indicators with larger spatial variability for SH evaluation at different scales, such as the relatively stable factors at the macro scale, including slope and aspect, thickness of the soil layer, and soil barrier layer [92,93]. For county and field scales, the relatively variable factors such as soil organic matter, soil nutrient elements, soil aggregates, soil respiration, and soil microorganisms are taken into account [94,95]. It is worth noting that soil organisms directly participate in soil processes and profoundly affect soil ecosystem services. Therefore, biological indicators are considered to be more sensitive to soil management. However, there are also some problems associated with biological indicators, including the representativeness of the collected samples, measurement costs, and relevance of SFs. In addition to the commonly used indicators such as microbial carbon and nitrogen, mineralizable nitrogen, soil biological indicators, soil biodiversity, functional gene expression, and some indicator organisms (nematodes, earthworms, etc.) have also been widely applied to analyze SH. Thus, development of convenient, fast, and low-cost tools or technologies to observe the changes in the characteristics of SH is crucial to achieve SH management and protection [1].

Evaluation indicators and magnitude of different scales are not fixed but are adjusted based on temporal and spatial change characteristics of the evaluation indicator and purpose of soil management. In this article, the study area is equivalent to mesoscale, and the indicator system selection and division are relatively detailed. Nevertheless, owing to the limitation of data acquisition, the indicator system employed in the present study did not take into account important indicators such as microbial diversity [96,97]. As China has a wide variety of soil types and crops, complex topography, and climatic factors in different regions, coupled with differences in soil management measures and management objectives, it is difficult to uniformly define the standards for healthy soil [1]. Therefore, it is necessary to further adjust or refine the index system and classification scheme according to the regional soil ecological environment to better identify the obstacles to regional SH. Furthermore, it is possible to select representative plots under different soil types, land use types, ecological environments, and soil management scenarios to reveal the law of spatial differentiation and evolution of SH at different scales through controlled variable experiments to simulate the spatial differentiation characteristics of SH at different scales [98]. Moreover, the minimum indicator set for SH evaluation in different regions can be established [99], and a gradual exploration of SH, combined with crop health and human health, can be used to investigate the influence of SH on human health [100]. Human productivities are inseparable from agricultural activities and soil. Scientific, reasonable, and sustainable field management is one of the core goals of maintaining human destiny and SH. For effective soil management, a new generation of managers, farmers, policymakers, and scientists who will understand the importance of soil system is necessary. To achieve this objective, education programs, starting at primary-school level, are essential. It is crucial to educate people to recognize the importance of soil for SFs and, ultimately, ecosystem services that are vital for all, and promote organic

farming, mulching, and minimum or zero tillage [101]. Besides, farmers should be made aware of the possibility of choosing higher quality products and simultaneously reducing environmental pollution with agrochemicals.

With regard to the application of SH evaluation, the two farmland resource quality evaluation systems formulated by the Ministry of Natural Resources and Ministry of Agriculture and Rural Affairs provide a solid foundation for the utilization and management of farmland. However, they mainly represent the "background" production potential of the soil, help in the cultivation of farmland and soil fertility, and do not adequately consider SF and soil environmental conditions. As a result, the evaluation results have a single effect and may not support the current multi-target cultivated SH management and protection. Among these systems, "Cultivated Land Quality Grade" (GB/T 33469-2016) uses difficult-to-quantify cleanliness and biodiversity indicators to characterize SH. The degree of cleanliness reflects the extent to which the soil is affected by toxic and harmful substances such as heavy metals, pesticides, and agricultural film residues. Biodiversity reflects the biological activity of soil ecosystems. The existing SH assessment system in China mainly focuses on soil pollution and lacks a comprehensive and standard tool. The German Müncheberg SQ evaluation system has been successfully piloted in some countries [102,103], and the principles and concepts of Müncheberg evaluation plan design can be applied to SH evaluation based on different SFs in China. Liu et al. [76] proposed a new classification system for evaluating farmland productivity and environmental quality in China based on the German Müncheberg SQ evaluation system, considering the concept of functional soil management and five SFs related to agriculture and forestry production and a variety of soil stresses. In their proposed system, each SF selects basic indicators from meteorological, topographic, hydrological, and soil physical and chemical properties. The stress factors of natural environment and human activities that affect SH determine the hazard indicators. According to the inherent and dynamic properties of the basic indicators, two sets of hazard indicator multiplier factors are designed. The SH grade is diagnosed with the product of the weighted score of basic indicators and multiplier of hazard indicators. However, under different anthropogenic climate conditions, the multiplicative factors in the hazard indicators are subjective for a specific SF. Therefore, in the future, it is necessary to reconsider the multiplicative factors according to soil characteristics, environment, land use, and soil management practices for different land ecosystems. In the present study, given the difficulty of data acquisition at county (regional) scale, we only discussed some limitations of our study dataset and comprehensively analyzed how the evidence supports our research hypotheses.

## 5. Conclusions

A new SH evaluation system based on the concept of functional soil management was developed in this study. The analytic hierarchy process and coefficient of variation method were utilized to determine the weights of the inherent and dynamic attribute indicators. Based on relevant literature and expert knowledge on the standardization of the evaluation indicators, a limiting factor multiplier was established according to the impact of hazard indicators on intrinsic and dynamic attribute indicators to evaluate the farmland SH of the study area. It not only takes into account the information carried by original data of the indicators featured by strong objectivity, but also increases the empirical knowledge of decision-makers and experts so that the results of SH evaluation conform to the actual situation to a certain extent. The developed evaluation method follows the principle of least limiting factor, which can reveal the nature of SH and SQ, as well as highlight the restrictive effects of hazards or obstacles on SH. This concept of SH evaluation based on functional soil management enables one to both objectively describe the various functions of soil and evaluate SH in a targeted manner, in line with the scientific and practical requirements of agricultural land utilization and soil management and protection. In the future, a variety of comprehensive evaluation methods can be developed, and their results can be compared to determine the best evaluation plan.

Based on the developed SH evaluation model, a digital soil assessment framework was utilized in this study to produce a map of SH and five SFs for Yixing City. A well-defined spatial model can help to determine the key characteristics of SF and SH as well as obstacles to SH. Healthy soils were predominant in Heqiao Town, Gaocheng Town, Zhoutie Town, and Wanshi Town in the northeast of Yixing City; sub-healthy soils were primarily distributed in Xushe Town in the mid-west and Guanlin Town, Gaocheng Town, and Zhoutie Town in the northeast; and unhealthy soils were scattered in Xushe Town and Dingshu Town in the mid-west, Zhangzhu Town and Hufu Town in the mid-south, Xinjie Street in the central part, and Xushe Town in the mid-west. The application of the obstacle degree model to diagnose the obstacle factors affecting SH of the study area can be crucial for targeted SH management and protection efforts in the future. The distribution of the obstacle degree of SH evaluation indicators, especially CEC, nutrient elements, and soluble carbon, was noted to be extremely different from that of the SH evaluation index obstacles in Yixing City. According to SH evaluation based on functional soil management, the obtained results provide a clear direction for scientific soil management and improvement, and the developed evaluation method can be applied in China and other parts of the world. Quantitative SH evaluation can help to understand the factors affecting SH of a complex soil system, and it is crucial to develop effective strategies for the sustainable utilization of agricultural land.

**Author Contributions:** Conceptualization, R.Z. and K.W.; methodology, R.Z.; validation, R.Z.; formal analysis, R.Z.; investigation, R.Z.; resources, K.W.; data curation, R.Z.; writing—original draft preparation, R.Z.; writing review and editing, K.W.; visualization, R.Z.; supervision, K.W.; project administration, K.W.; funding acquisition, K.W. All authors have read and agreed to the published version of the manuscript.

**Funding:** This study was supported by the National Key R&D Program of China (No. 2018YFE0107000).

**Institutional Review Board Statement:** Not applicable.

**Informed Consent Statement:** Not applicable.

**Data Availability Statement:** The datasets used and/or analyzed during the current study are available from the corresponding author upon reasonable request.

**Acknowledgments:** Thank you to everyone who contributed to this study.

**Conflicts of Interest:** The authors declare no conflict of interest.

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
