# Peer review of "Soil Health Evaluation of Farmland Based on Functional Soil Management—A Case Study of Yixing City, Jiangsu Province, China"

_agriculture, doi:10.3390/agriculture11070583_

Round 1

Reviewer 1 Report

The paper was improved as suggested 

Reviewer 2 Report

About this version of the article i donr'have any suggestion for authors

This manuscript is a resubmission of an earlier submission. The following is a list of the peer review reports and author responses from that submission.

Round 1

Reviewer 1 Report

Interesting to see work in China on Soil Health, a topic that receives increasing attention worldwide. I recommend that you include some recent work elsewhere to update your paper.. Also clarification is needed on several issues that are outlined in my attached detailed comments.  

Reviewer 2 Report

Dear author, I suggest to reorganize the work. The scientific objctives are not very clear, the introduction is too long and dispersive. Moreover, the indicators are not cleary showed. Authors should explaine why some indicators missing, such as C org and microbial carbon. Tables  are too much and not .  I suggest to reorganize the work focusing the  aim of work on the presentation of data collected and the relative discussion.
